# Effect of Ferulic Acid and Its Derivatives on Cold-Pressed Flaxseed Oil Oxidative Stability and Bioactive Compounds Retention during Oxidation

**DOI:** 10.3390/foods12051088

**Published:** 2023-03-03

**Authors:** Natalia Mikołajczak, Wojciech Pilarski, Krzysztof Gęsiński, Małgorzata Tańska

**Affiliations:** 1Department of Food Plant Chemistry and Processing, Faculty of Food Sciences, University of Warmia and Mazury in Olsztyn, 10-718 Olsztyn, Poland; 2Department of Biology and Plant Protection, Faculty of Agriculture and Biotechnology, Bydgoszcz University of Science and Technology, 85-796 Bydgoszcz, Poland

**Keywords:** vanillic acid, dihydroferulic acid, 4-vinylguaiacol, Rancimat test, fatty acids, quality indices, bioactive compounds

## Abstract

Ferulic acid (FA) is a naturally occurring phenolic antioxidant that is widely used in the food, pharmaceutical, and cosmetic industries due to its low toxicity. Its derivatives also find numerous industrial applications and may have even higher biological activity than ferulic acid. In this study, the effect of the addition of FA and its derivatives—including vanillic acid (VA), dihydroferulic acid (DHFA), and 4-vinylguaiacol (4-VG)—on the oxidative stability of cold-pressed flaxseed oil and the degradation of bioactive compounds during oxidation was investigated. The results showed that FA and its derivatives affected the oxidative stability of flaxseed oil, but their antioxidant activity depended on the concentration (25–200 mg/100 g oil) and temperature of treatment (60–110 °C). Based on Rancimat test results, flaxseed oil oxidative stability predicted at 20 °C increased linearly with ferulic acid concentration, while its derivatives effectively prolonged the induction time at lower concentrations (50–100 mg/100 g oil). The addition of phenolic antioxidants (80 mg/100 g) generally showed a protective effect against polyunsaturated fatty acids (DHFA and 4-VG), sterols (4-VG), tocols (DHFA), squalene, and carotenoids (FA). The exception was VA, which increased the degradation of most bioactive compounds. It is believed that adding properly composed mixtures of FA and its derivatives (DHFA and 4-VG) can extend the shelf life of flaxseed oil and provide nutritional benefits.

## 1. Introduction

Flaxseed (linseed) oil is one of the most important vegetable oils in the world and is commonly produced by mechanical pressing or solvent extraction. The cold-pressing technique has attracted considerable interest in the last decade due to its capability of producing a high-quality product as well as its lower energy requirements, ease of operation, and environmentally friendly approach. The maximum temperature of cold-pressed oil should not exceed 50 °C, and only its physical purification through filtration, sedimentation, or centrifugation processes is allowed [1]. Flaxseed oil is characterized by a high content of unsaturated fatty acids (UFA). It has been reported in the literature that their contribution to total fatty acids can be as high as 92% [2,3], with the majority (49–66%) consisting of α-linolenic acid (ALA) [3,4,5,6]. The high content of ALA makes flaxseed oil a valuable preventive and medicinal product, e.g., for cardiovascular and nervous system diseases and cancers [6,7,8]. Unfortunately, the presence of three double bonds in the ALA structure makes the oil very susceptible to the oxidation process [9,10]. It is estimated that its shelf life ranges from 5 weeks to 3 months when stored refrigerated [4,9].

The protection of vegetable oils, in particular oils with a high content of UFA, is achieved through the addition of antioxidants that can interact with oxidation products (free radicals) through their ability to scavenge or decompose them [7,10]. In recent years, phenolic compounds have attracted considerable attention due to their beneficial antioxidant activity [11,12]. For example, Michotte et al. [13] investigated the possibility of using flavonols (myricetin, catechin) and hydroxycinnamic acid (caffeic acid) as additives to increase the stability of flaxseed oil, and in studies by Suja et al. [14], the addition of lignans was used to reduce the negative effects of oxidation of soybean, sunflower, and safflower oils, while Mikołajczak et al. [15] showed that vinyl phenolic acid derivatives have protective effects on the fatty acids and bioactive compounds in flaxseed and rapeseed oils. A new approach to potentially increase the oxidative stability of edible oils is the use of antioxidants naturally occurring in foods, which may provide additional health benefits to consumers [16]. One such antioxidant could be ferulic acid (FA), which has low toxicity, is more easily absorbed by the body, and stays in the blood for longer periods than other phenolic acids [17].

FA (4-hydroxy-3-methoxycinnamic acid) is an important phenolic acid commonly found in the leaves, fruits, and seeds of plants [18]. One of the best documented biological activities of FA is its antioxidant properties, which are mainly due to the presence in the molecule of the phenol nucleus and an extended side chain [17,19,20]. This helps FA easily form a stable phenoxy radical and provides high protection against adverse oxidation processes in DNA, lipids [17,18,21], and proteins [22,23]. FA may also have beneficial effects in the prevention or treatment of oxidative stress disorders, i.e., cardiovascular diseases, cancers, and diabetes [19,21].

The shikimic pathway produces FA in plants starting from aromatic amino acids such as L-phenylalanine and L-tyrosine [20,24], which are initially converted to p-coumaric acid and cinnamic acid using phenylalanine ammonium lyase and tyrosine ammonium lyase, respectively. p-Coumaric acid is converted to FA through hydroxylation and methylation reactions [18]. Metabolic studies have shown that FA can also be metabolized in vivo, resulting in several metabolites such as FA-glucuronide, FA-sulfoglucuronide, ferulic acid sulfate (major metabolites in rat plasma and urine), FA-diglucuronide, m-hydroxyphenylpropionic acid, feruloylglycine, dihydroferulic acid (DHFA), vanillic acid (VA), and vanillylglycine [20,25,26]. These data suggest that the main FA metabolic pathway is glucuronidation and/or sulfate conjugation [18]. FA conjugation occurs in the liver using sulfur transferases and glucuronosyl transferases of uridine diphosphate (UDP); the conjugation reaction may also occur partly in the intestinal mucosa and kidney [25,26]. Most likely, free FA is metabolized in the liver via β-oxidation [18,21]. The literature also reports that FA is transformed by numerous fungi, actinomycetes, and yeasts into various useful organic compounds [27,28]. As a result of the bioconversion process (non-oxidative decarboxylation), FA is transformed into 4-vinylguaiacol (4-VG) due to the decarboxylase enzyme [27]. It can also be degraded to vanillin, among others, via the protocatechuate 4,5-cleavage pathway [18,27]. Both vanillin and 4-VG are considered value-added bioproducts and are widely used in food and cosmetics production [28]. Furthermore, it has been proven that FA is a powerful food preservative because of its antimicrobial and antioxidant activities [18]. 

Although more studies are being conducted to determine the effects of phenolic compounds on oil oxidative stability, few researchers have used FA and its derivatives to stabilize ALA-rich oils [15,29,30]. It is emphasized that natural FA is easily available and its cost is relatively low, but the applications of FA in food are limited by its low hydrophobicity, hydrophilicity, and stability in various solvent systems [31]. However, FA is a highly reactive compound that shows high potential for the preparation of its derivatives with higher solubility in water and/or higher biological activity [32,33]. 

In this study, the antioxidant actions of different concentrations of FA and its derivatives (VA, DHFA, and 4-VG) in cold-pressed flaxseed oil were compared at different temperatures using the Rancimat test. Additionally, the effect of added phenolic additives on the degradation of bioactive compounds (unsaturated fatty acids, sterols, tocols, squalene, and carotenoids) under oxidation at 60 °C was evaluated.

## 2. Materials and Methods

### 2.1. Reagents

Analytical-grade reagents and solvents, i.e., Folin–Ciocalteu reagent from Sigma-Aldrich (Saint Louis, MO, USA), anhydrous sodium sulfate (Na_2_SO_4_), chloroform, diethyl ether, ethanol (99.9% purity), formic acid, methanol, potassium hydroxide (KOH), sodium carbonate (Na_2_CO_3_), sulfuric acid (H_2_SO_4_), and zinc purchased from Chempur (Piekary Śląskie, Poland) were used. Chromatography-grade solvents, i.e., dichloromethane, ethanol, methanol, n-hexane, heptane, methyl tert-butyl ether, iso-propanol, pyridine, and N,O-bis(trimethylsilyl)trifluoroacetamide (BSTFA) with 1% trimethylchlorosilane (TMCS) were purchased from Sigma-Aldrich (Saint Louis, MO, USA). Analytical standards such as 5α-cholestane, margaric acid, β-apo-8’-carotenal, D-catechin, phenolic acids (p-coumaric acid (p-CA), dihydroferulic acid (DHFA), sinapic acid (SA), vanillic acid (VA)), and 4-vinylguaiacol (4-VG) were purchased from Sigma-Aldrich (Saint Louis, MO, USA), while tocopherols (α-, γ-, and δ-tocopherol) were purchased from Calbiochem (Nottingham, UK). Deionized water was obtained from HLP 5 deionizer (Hydrolab, Gdańsk, Poland). 

### 2.2. Materials

FA and its derivatives (VA, DHFA, and 4-VG) used in the study were bought as analytical standards with a declared purity > 95%.

The research material was commercial cold-pressed flaxseed oil purchased from “Olejarnia Świecie” (Świecie, Poland) and obtained immediately after its production. The oil was analyzed after opening the package and stored at −20 °C between analyses. Water content in fresh oil was 0.06% (determined by the Karl Fischer method using a 917 Coulometer set equipped with a diaphragm (Methrom, Herisau, Switzerland)).

### 2.3. Accelerated Oxidation Test

Mixtures of cold-pressed flaxseed oil with FA and its derivatives were prepared as follows: FA and its derivatives (VA, DHFA, and 4-VG) were dissolved in ethanol and added to oil samples (100 g) to obtain a final concentration of 25, 50, 75, 100, 150, and 200 mg per 100 g of oil. An accelerated oxidation test was evaluated on a Rancimat apparatus 743 (Metrohm, Herisau, Switzerland) at 60, 80, and 110 °C with an air flow rate of 20 L/h according to AOCS Official Method Cd 12b–92 [34]. The time that elapsed until these oxidation products appeared was saved as the induction time (IT).

Oxidized oil and oxidized oils with phenolic additives were obtained using an accelerated oxidation test at 60 °C for 10 days in a thermal research chamber (KBC-100 W type; WAMED, Warsaw, Poland) according to AOCS Official Method Cg 5–97 [35]. The oil mixtures were prepared as before, but FA and its derivatives were added to the oil sample (200 g) to obtain a final concentration of 80 mg per 100 g of oil. In the thermostatic test, the oils were stored in closed bottles made of dark glass (with a volume of 250 mL).

### 2.4. Determination of Quality Indices and Fatty Acid Composition

The quality indices of oils included acid (AV), peroxide (PV), and anisidine (AnV) values, and the contents of conjugated dienes and trienes were determined on the basis of the following AOCS Official Methods: Te 1a–64 [36], Cd 8b–90 [37], Cd 18–90 [38], and Ch 5–91 [39], respectively.

The fatty acid composition of oils was determined according to the procedure described by Mikołajczak et al. [40]. Methyl-esters of fatty acids were prepared as follows: the oil sample with an internal standard (margaric acid) was methylated at 70 °C for 2 h in a methylating mixture (methanol:chloroform:sulphuric acid 100:100:1, *v*/*v*/*v*). After neutralizing the H_2_SO_4_ through the addition of zinc, the solvent was evaporated under a stream of nitrogen. The residue was dissolved in n-hexane and analyzed using a GC-MS QP2010 PLUS gas chromatograph (Shimadzu, Tokyo, Japan) with an SGE BPX-70 capillary column (25 m × 22 mm × 0.25 μm, SGE Analytical Science, Ringwood, Australia) against helium (carrier gas) applied with a flow rate of 1.3 mL/min. The GC-MS conditions were as follows: the interface temperature was set at 240 °C, the ion source temperature was set at 240 °C, and the column temperature was programmed in the range of 150–250 °C. The electron energy was set at 70 eV. Fatty acids were identified based on mass spectra, and their contents were calculated with reference to the internal standard. The repeatability for determining margaric acid was 2.5% (expressed as a coefficient of variation), and the limit of quantification (LOQ) was 0.05 μg/g of oil.

### 2.5. Determination of Squalene and Sterol Contents

The content of sterols and squalene was determined according to the procedure described by Mikołajczak et al. [40]. The compounds were extracted from the oils as follows: α-cholestane (internal standard) solution in ethanol and 2 M KOH solution in ethanol were added to the sample. The mixture was heated at 70 °C for 30 min. The non-saponifying fraction was extracted three times with diethyl ether, and the collected extracts were rinsed with deionized water. The solvent was evaporated on a Büchi R-210 rotary evaporator (BÜCHI Labortechnik AG, Flawil, Switzerland) at 45 °C. Derivatization was carried out as follows: pyridine and BSTFA with 1% TMCS were added to the dry extract, and then the mixture was heated at 60 °C for 1 h. Next, the solution was diluted in heptane and analyzed using the GC-MS QP2010 PLUS chromatograph (Shimadzu, Tokyo, Japan) coupled with a mass spectrometer and equipped with a ZB-5MSi capillary column (30 m × 0.25 mm × 0.25 μm, Phenomenex, Torrance, CA, USA). Helium was applied as a carrier gas with a flow rate of 0.9 mL/min. The GC-MS conditions were as follows: injector temperature was set at 230 °C, column temperature was set at 70 °C for 2 min, increased to 230 °C at the rate of 15 °C/min, and to 310 °C at the rate of 3 °C/min, and maintained for 10 min, GC-MS interface temperature was set at 240 °C, ion source temperature was set at 220 °C, and electron energy was set at 70 eV. The total ion current (TIC) mode was used for quantification (100–600 *m*/*z* range). Sterols and squalene were identified based on retention times and mass spectra, and their contents were calculated with reference to the internal standard. The repeatability for determining α-cholestane was 2.5% (expressed as a coefficient of variation), and the LOQ was 0.05 μg/g of oil.

### 2.6. Determination of Tocol Contents

The content of tocols was determined according to the procedure described by Mikołajczak et al. [40]. The oil sample was diluted in n-hexane and centrifuged on an Eppendorf Centrifuge 5417R type (Eppendorf AG, Hamburg, Germany) for 10 min (16,000 rpm). The resultant solution was analyzed using an HPLC Agilent Technologies 1200 chromatograph (Santa Clara, CA, USA) equipped with a fluorescence detector of the same company and a LiChrospher Si 60 column (250 mm × 4 mm × 5 μm, Merck, Darmstadt, Germany). A 0.7% solution of iso-propanol in n-hexane was used as the mobile phase at a flow rate of 1 mL/min. The fluorescence detector was set at excitation and emission wavelengths of 296 nm and 330 nm, respectively. The content of tocols was determined from calibration curves prepared for tocopherol standards. The repeatability for determining tocopherol contents was 2.5% (coefficient of variation). The LOQs were 0.45, 0.4, and 0.2 μg/g of the sample for α-, γ-, and δ-tocopherol, respectively. The linearity of the calibration curves was confirmed in the range of 0.02–16 mg/L.

### 2.7. Determination of Carotenoid Content

The content of carotenoids was prepared and analyzed according to the methodology given by Mikołajczak et al. [40]. After being diluted in n-hexane with β-apo-8’-carotenal (internal standard), the oil sample was saponified by adding a 40% KOH solution in methanol. The solution was shaken in the Multi Rotator RS-60 (Biosan, Riga, Latvia) in the dark at room temperature for 16 h. After saponification, 10% Na_2_SO_4_ was added to the sample, and the extraction of carotenoids was carried out four times with n-hexane. The collected extract was evaporated on a Büchi R-210 rotary evaporator at 225 mbar and 45 °C, and the dry extract was dissolved in a methanol:dichloromethane mixture (45:55, *v*/*v*). The resultant solution was analyzed by RP-HPLC technique using an Agilent Technologies 1200 chromatograph, equipped with a diode array detector (DAD), a YMC-C30 chromatography column (150 mm × 4.6 mm × 5 μm) and a YMC-C30 precolumn (10 mm × 4.6 mm × 3 μm) (YMC-Europe GmbH, Dinslaken, Germany). The column temperature was 30 °C and a methanol (A) and methyl tert-butyl ether (B) gradient was used for elution. The gradient was programmed as follows: 0–5 min 95% of A, until 25 min decreases to 72% of A, and keeps decreasing until 33 min to 5% of A. Next, it increases to 95% of A and is stable until 60 min. Absorbance was measured at 450 nm. Carotenoids were identified based on the retention times and absorption spectra of the individual carotenoids, and their contents were calculated with reference to the internal standard. The repeatability for determining β-apo-8’-carotenal content was 2.5% (coefficient of variation). The LOQ was 0.05 μg/g of the sample, while the linearity of the calibration curve was confirmed in the range of 1–150 mg/L.

### 2.8. Determination of Phenolic Compounds

The total content of phenolic compounds was determined spectrophotometrically by color reaction according to the procedure described by Mikołajczak and Tańska [3]. Phenolic compounds from the oil sample were extracted three times with 80% methanol (1:10, *w*/*v*). The collected extracts were evaporated under a vacuum using a Büchi R-210 rotary evaporator. A solution of Folin-Ciocalteu reagent in water (1:2, *v*/*v*), 14% Na_2_CO_3_ solution, and deionized water was added to the dry residue. The mixture was incubated in a dark place at room temperature for 1 h. The absorbance of the solution was measured using a microplate reader (FLUOstar OMEGA; BMG Labtech GmbH, Ortenberg, Germany). The phenolic compound content was calculated on the basis of the calibration curve prepared for D-catechin.

The extraction of phenolic acids was performed using a vacuum equalizer (Witko, Łódź, Poland) and SPE Supelco columns filled with the diol (500 mg) (Sigma-Aldrich, Saint Louis, MO, USA), as described by Mikołajczak et al. [15]. The phenolic acids were analyzed by RP-HPLC technique using an Agilent Technologies 1200 chromatograph, equipped with a photodiode detector and Eclipse XDB-C18 chromatography column (150 mm × 4.6 mm × 5 μm, Agilent Technologies, Santa Clara, CA, USA). The column temperature was 30 °C, and the mobile phase consisted of water:acetonitrile:formic acid (88:10:2, *v*/*v*/*v*). The isocratic flow rate was equal to 0.8 mL/min. The detection was performed at the wavelengths of 260 and 320 nm. Phenolic acids were identified based on the retention times and absorption spectra of the individual phenolic acids. The content of phenolic acids was determined from calibration curves prepared for phenolic acid standards (FA, p-CA, DHFA, SA, VA, and 4-VG). The repeatability for determining phenolic acid contents was at least 2.1% (coefficient of variation). The LOQ was at least 0.05 μg/g of the sample, while the linearity of calibration curves was confirmed in the range of 1–150 mg/L.

### 2.9. Statistical Analysis

All analyses were performed in triplicate, and the obtained results were analyzed using Statistica 13.1 PL software (StatSoft, Cracow, Poland). The differences between the samples were determined using a one-way analysis of variance (ANOVA), followed by a Tukey test at a *p* ≤ 0.05 significance level.

## 3. Results and Discussion

### 3.1. Effect of Different Concentrations of Ferulic Acid and Its Derivatives on Oxidative Stability of Cold-Pressed Flaxseed at Different Temperatures 

In Figure 1, the changes are shown in the IT of cold-pressed flaxseed oil mixtures with FA and its derivatives, i.e., VA, DHFA, and 4-VG. The phenolic additives were added in amounts of 25–200 mg per 100 g of oil, and their antioxidative effect was evaluated in the Rancimat test at 110 °C (Figure 1a), 80 °C (Figure 1b), and 60 °C (Figure 1c).

The use of an increasing amount of FA in the Rancimat test at 110 °C resulted in a linear increase in the IT of cold-pressed flaxseed oil (Figure 1a). It was determined that the parameter reached up to 2.83 h for 200 mg of FA per 100 g of oil. VA addition at a concentration of 25 mg/100 g showed the most beneficial effects (2.65 h), and its higher concentration in oil (50–200 mg/100 g) resulted in a gradual decrease in IT. In turn, the antioxidant activities of both DHFA and 4-VG were the best at a concentration of 75 mg/100 g. The most noticeable changes in IT of cold-pressed flaxseed oil were after the addition of 4-VG in an amount of 75 mg/100 g, where IT was 3.52 h (increased by 40.8% compared to the control sample).

Lowering the temperature in the Rancimat test to 80 °C resulted in an almost 10-fold increase in the IT of cold-pressed flaxseed oil (Figure 1b). IT elongated proportionally (as concentration increased) when FA was added. The FA addition at a concentration of 200 mg/100 g increased IT up to 30.62 h (over a 19% increase compared to the control sample). In turn, with the addition of VA, the IT of the oil gradually decreased. It was noted that the control oil had an IT of about 26 h, while after the addition of VA at 200 mg/100 g, it was more than 8% lower. DHFA addition improved IT the most at a concentration of 75 mg/100 g; at the higher concentrations, the elongation of IT was still observed but at a lower rate. The addition of 4-VG at a concentration of 25–100 mg/100 g resulted in a reduction of oil IT by 2–9% (depending on the concentration).

Results of the IT in the Rancimat test at 60 °C showed that the phenolic additives generally improved the oxidative stability of cold-pressed flaxseed oil, but their effectiveness varied (Figure 1c). A linear relationship was observed between IT values and FA concentration. The highest amount of this phenolic additive resulted in an increase in IT of up to 60% compared to the control sample. DHFA exhibited lower antioxidant effectiveness and caused a noticeable elongation of IT (an increase of 14–42%) at a concentration of 25–100 mg/100 g, but a reverse relationship was discovered. The addition of 4-VG increased IT up to 124 h at a concentration of 75 mg/100 g, but its additions up to 150 mg per 100 g of oil showed comparable antioxidant activity at 60 °C. The effect of VA on the antioxidant stability of flaxseed oil was independent of its concentration in the oil. The phenolic additive application increased IT by 10–18% compared to the control sample. 

Based on the Rancimat tests at different temperatures, the shelf life of the cold-pressed flaxseed oil at ambient conditions (20 °C) was predicted, and the results are presented in Table 1. The values of this parameter were calculated by plotting the logarithms of ITs versus elevated temperatures and extrapolating them to room temperature [41]. It was evaluated that flaxseed oil with the lowest FA addition (25 mg/100 g) can be stored at 20 °C for 3.71 months, while the control oil (without additives) can be stored for 2.47 months. The higher concentrations of FA in the flaxseed oil extended its shelf life up to 5.31 months (200 mg/100 g). The addition of VA increased the shelf life of flaxseed oil by 23–34% when compared to the control sample, but the observed changes were independent of the additive concentration in the oil. The beneficial effects of DHFA addition were noticeable only at its lower concentrations in oil, ranging from 25 to 75 mg/100 g. In the case of 4-VG, the flaxseed oil shelf life was improved when 100 g of the oil was combined with 75–150 mg of this ferulic acid derivative.

It is worth mentioning that cold-pressed flaxseed oil is a widely studied edible oil, and available results of its oxidative stability are varied. In the work of Bozan and Tamelli [42], IT of flaxseed oil at 110 °C was 1.57 h. In contrast, the IT of flaxseed oils analyzed by Tańska et al. [9] was in the range of 2.00–4.33 h. The oxidative stability of cold-pressed flaxseed oils was also evaluated in the Rancimat test at 100 °C. For example, Raczyk et al. [43] showed that the IT of cold-pressed flaxseed oil was from 3.47 to 5.63 h. Similar results were presented by El-Waseif et al. [44] and Symoniuk et al. [45] with the same measured parameters (ca. 4.90 h). Symoniuk et al. [45] analyzed the stability of cold-pressed flaxseed oils in the Rancimat test at various temperatures (70–140 °C). The authors showed that the IT of the tested oils decreased with increasing temperature and was, on average, 36.81 h at 70 °C and 0.26 h at 130 °C.

Furthermore, the oxidative stability of flaxseed oil was improved by adding different antioxidants, including various phenolic compounds. Tańska et al. [46] analyzed the effect of phenolic acid derivatives (4-vinylsyringol (4-VS) and 4-VG) on the stability of three commercially available oils, including cold-pressed flaxseed oil. Phenolic additives were added at concentrations of 20, 40, and 80 mg per 100 g oil and oxidized in the Rancimat test at 110 °C. In general, 4-VG was more effective; the increase in IT after its addition was 5 to 25-fold greater than that of 4-VS. The highest increase was recorded in the case of cold-pressed flaxseed oil, for which the addition of 80 mg 4-VG per 100 g of oil resulted in a 50% increase in IT. The authors suggest that antioxidant activities depend not only on the total phenol content, but also on the type of phenolics present [47]. Cinnamic acid derivatives have high antioxidant activity, which results from the presence of vinyl fragments. However, the reactive center (vinyl fragment) is significantly influenced by the substituent present in various positions of the benzene nucleus [48]. Karamać et al. [49] determined that the activity of phenolic compounds depends on the hydroxyl number of the moieties attached to the aromatic phenol ring, which indicates dihydroxyphenolic acids are more active than their monohydroxy counterparts. In addition, the presence of an ethylene side chain containing an unsaturated bond increases the ability to transfer electrons, stabilizes the resulting phenoxy radical, or offers an additional place to react with reactive oxygen species (ROS) [50].

The temperature factor also seems to affect the antioxidant properties of phenolic compounds. Temperature fluctuations can change the mechanism of action of some antioxidants and affect them in different ways or affect certain reactions in which antioxidants are involved (mainly reactions with lipid radicals) [51]. Antioxidants can also evaporate from the matrix at much higher temperatures [52]. The literature indicates that there is a linear relationship between temperature and the antioxidant activity of phenolic compounds, mainly phenolic acids. For example, Réblová [51] discovered a decrease in antioxidant activity with increasing temperature for VA, similar to our study. Marinova and Yanishlieva [53] showed that an increase in temperature does not change the antioxidant activity of benzoic acid derivatives, while the activity of cinnamic acid derivatives increases. The literature also suggests that the antioxidant activity of FA is dependent on its solubility in oil. This is particularly noticeable in the pharmaceutical industry, where FA is used in skincare and cosmetic applications, making it difficult to formulate oil-based formulations. Therefore, various methods are sought to increase its solubility. An interesting approach is the development of lipophilic feruloylated derivatives through transesterification with different solvents, i.e., ionic liquids, and supercritical carbon dioxide [32,33,54].

### 3.2. Changes in Quality of Cold-Pressed Flaxseed Oil with the Addition of Ferulic Acid and Its Derivatives in an Oxidation Test

Table 2 presents quality indices of fresh and oxidized flaxseed oil samples. AV of fresh cold-pressed flaxseed oil was 1.39 mg KOH/100 g, while PV and AnV values were at a low level (0.39 mEq O_2_/kg and 0.61, respectively). In the flaxseed oil, no trienes were found, while the content of dienes reached 0.18%. 

Heating cold-pressed flaxseed oil at 60 °C accelerated the hydrolysis and oxidation processes (Table 2). The AV increased by almost 10% compared to fresh oil. The use of FA and VA additives resulted in an increased amount of free fatty acids in cold-pressed flaxseed oil (AV = 1.67 mg KOH/g). Only the addition of 4-VG resulted in a reduction of AV by more than 40%. The PV in heated oil samples ranged from 6.30 to 9.88 mEq O_2_/kg, while oxidized control oil and oxidized oil with the addition of FA were characterized by the lowest values of the quality parameter. The 4-VG showed an unfavorable effect because the oil with its addition was characterized by an even higher content of primary oxidation products than the oil without additives (PV was almost 1.5 times higher). Oil with the addition of VA had the lowest content of secondary oxidation products (AnV = 2.60). The AnV in the other oils was comparable to each other and to the oil without any additives (AnV = 14.33). The oxidation process caused an almost 2-fold increase in the share of dienes (0.26–0.30%), while the share of trienes remained unchanged (0.00% in all analyzed oil samples).

Although the hydrolysis and oxidation rates in flaxseed oil after its heating at 60 °C varied in terms of the type of phenolic additive, the AV and PV in all samples did not exceed the values indicated in the acceptable standards for cold-pressed oils [55], i.e., 4.0 mg KOH/g and 15.0 mEq O_2_/kg, respectively. Hasiewicz-Derkacz et al. [29] confirmed a more pronounced effect for FA compared to our study. They added 0.5 mM FA to cold-pressed flaxseed oil from transgenic seeds (ALA accounted for 2–20% of the total fatty acids) and reported an 85% decrease in the formation of oxidation products at 140 °C.

### 3.3. Changes in Fatty Acid Composition of Cold-Pressed Flaxseed Oil with the Addition of Ferulic Acid and Its Derivatives in an Oxidation Test

The total fatty acid content of fresh cold-pressed flaxseed oil was 97.88 g/100 g; more than 60% were polyunsaturated fatty acids (PUFA) (Table 3). Among them, α-linolenic acid (C18:3) predominated, and the content of linoleic acid (C18:2) was almost 5-fold lower. The analyzed cold-pressed flaxseed oil was also characterized by a low content of saturated fatty acids (SFA), in which the content of palmitic acid (C16:0) did not exceed 5 g/100 g and the content of stearic acid (C18:0) was more than 2-fold lower.

The oxidation process caused changes in the fatty acid composition of cold-pressed flaxseed oil (Table 3). The highest degree of their degradation was recorded in oil with the addition of VA (86.37 g/100 g), while the other phenolic additives contributed to the preservation of 94–97% of all fatty acids. The oxidation process itself also caused significant changes in the content of individual groups of fatty acids; their final contents in oil without phenolic additives were 5.45 g/100 g for SFA, 17.30 g/100 g for monounsaturated fatty acids (MUFA), and 68.65 g/100 g for PUFA. It was also noted that 4-VG significantly reduced SFA content by almost 15%. The most noticeable changes in the unsaturated fatty acids (MUFA and PUFA) were recorded in oil with the addition of VA (a decrease of 15.8% and 14.0% compared to fresh oil, respectively). In turn, the addition of DHFA revealed a protective effect on the unsaturated fatty acids (not statistically significant differences compared to fresh oil).

VA addition caused the largest changes in the content of individual fatty acids (Table 3). In oil with its addition, the content of oleic acid (C18:1) decreased by over 5%, the content of C18:2 acid by over 10%, and the content of C18:3 acid by almost 15%. It should be noted that the use of phenolic additives meant that most of the content of individual acids was preserved or their losses were small. However, phenolic additives in particular protected unsaturated fatty acids, e.g., DHFA and 4-VG reduced C18:3 acid oxidation (losses to 4%). Additionally, the C18:2 acid content was fully preserved by DHFA and 4-VG, while the C18:1 acid loss in oil with DHFA added was less than 2%.

Analysis of the composition of fatty acids indicates that the oil was pressed from flaxseeds, which are characterized by a high-fat content and a high percentage of α-linolenic acid from the omega-3 family [6]. A similar composition of fatty acids in cold-pressed flaxseed oil was found by Tańska et al. [3], Mikołajczak et al. [15], Lewinska et al. [5], and Zhang et al. [6].

### 3.4. Changes in the Content of Lipophilic Bioactive Compounds in Cold-Pressed Flaxseed Oils with the Addition of Ferulic Acid and Its Derivatives in an Oxidation Test

The fresh cold-pressed flaxseed oil used in the study was characterized by a squalene content of 4.43 mg/100 g (Table 4). The total sterol content was 630 mg/100 g, with β-sitosterol accounting for nearly 35% of the total sterols. The contents of cycloartenol and campesterol were predominant; there were more than 150 mg/100 g of each. Other sterols (stigmasterol and 25-hydroxy-24-methylcholesterol) together accounted for no more than 14% of total sterols. γ-Tocopherol predominated among tocols in cold-pressed flaxseed oil (68.18 mg/100 g). An equally high content was found for plastochromanol-8. The content of total tocols in flaxseed oil was 106.52 mg/100 g, with α-tocopherol accounting for only 5.76% of the total tocols in oil. Carotenoids were present in a low concentration (only 0.69 mg/100 g) in the studied flaxseed oil. The contents of β-carotene and lutein were dominant, and trace amounts of α-carotene and zeaxanthin were also found (up to 0.07 mg/100 g).

The greatest changes in squalene and total sterol contents were recorded in oil with the addition of VA (decreases of more than 50% and more than 8%, respectively) (Table 4). Furthermore, high losses of squalene were observed in oils with the addition of DHFA (over 46%) and 4-VG (over 43%). In these oils, losses were even higher than in oxidized oil without additives (the squalene content was 3.07 mg/100 g). Only the addition of FA resulted in the retention of almost 80% of the squalene content compared to the fresh oil. The total sterol contents in oils with additives such as FA and DHFA were 592.42 and 594.79 mg/100 g, respectively, and were similar to oxidized oil (586.81 mg/100 g). Individual sterols losses, such as campesterol, stigmasterol, β-sitosterol, 25-hydroxy-24-methylcholesterol, and cycloartenol, were comparable and did not exceed 10% in oils with FA and DHFA additions; they were also comparable to oxidized oil. The addition of VA to oil caused the greatest changes in the content of campesterol, stigmasterol, and β-sitosterol. The final contents of these compounds were 103.93, 29.45, and 197.62 mg/100 g, respectively. Moreover, the addition of VA significantly increased cycloartenol losses (which reached over 12%), even when compared to oxidized oil (153.38 mg/100 g). The content of cycloartenol in other oils ranged from 156.16 to 161.42 mg/100 g. The 4-VG addition showed protective properties for all sterol fractions; losses for their contents ranged from 1.43 to 3.31%. There were no significant differences in the content of unidentified sterol-like compounds in the oils with additives (46.41–47.00 mg/100 g), but they were higher than in oxidized oil without additives (46.23 mg/100 g). 

Total tocol contents in oils with additives ranged from 63.35 to 75.81 mg/100 g; the lowest content was observed in oil with 4-VG addition and the highest content in oil with DHFA addition (Table 4). The total tocol contents in oxidized oil and oils with the addition of FA and VA were over 70 mg per 100 g. α-Tocopherol has been completely degraded in oils, except for oil with the addition of DHFA, where over 50% of its content has been retained (compared to fresh oil). γ-Tocopherol losses were similar to each other and ranged from 24.01% (oil with VA) to 27.63% (oxidized oil). A significant decrease in its content was noted only in the oil with the addition of 4-VG, and the value was 44.83 mg/100 g. Furthermore, the content of plastochromanol-8 was significantly reduced (a decrease of over 42% compared to fresh oil) in oil with the addition of 4-VG. The content of plastochromanol-8 in oxidized oils with the addition of other phenolic additives was 21–22 mg/100 g.

The total content of carotenoids in cold-pressed flaxseed oil significantly decreased after the addition of 4-VG (a decrease of almost 74% compared to fresh oil) (Table 4). A high decrease in the content of carotenoids was also noted in oil with DHFA (a decrease of almost 50% compared to fresh oil). Carotenoid content changed similarly in oxidized oil and oil with VA (a decrease of almost 30% compared to fresh oil), whereas FA addition preserved nearly 80% of the total carotenoids present in fresh oil. The content of α-carotene in oxidized oil without additives and oil with DHFA was 0.02 mg/100 g; in oils with 4-VG, it decreased to 0.01 mg/100 g; and in oils with FA and VA, it was the highest (0.03 mg/100 g). The oxidation process significantly reduced the β-carotene content (by more than 50% compared to fresh oil) but adding 4-VG accelerated this process (by more than 80% compared to fresh oil). Only FA prevented oxidation of β-carotene; more than 95% was preserved. The used additions significantly reduced the lutein content in cold-pressed flaxseed oil (losses from 33.95 to 67.70%) compared to oxidized oil, but they did not affect the content of zeaxanthin.

Literature data indicate that the squalene and total sterol contents of cold-pressed flaxseed oils are varied. Mikołajczak et al. [15] stated that the squalene content is 2.39 mg/100 g, and the total sterol content reaches almost 285 mg/100 g. Tańska et al. [3] determined that the squalene content in commercial cold-pressed flaxseed oils is in the range of 1.01–4.29 mg/100 g and the total sterol content ranges from 409.40 to 538.83 mg/100 g. Many authors [3,7,15,56] reported that the main sterol fractions in cold-pressed flaxseed oil are also β-sitosterol, cycloartenol, and campesterol. According to research conducted by Tańska et al. [3], the total tocol content in cold-pressed flaxseed oil ranges from 48.88 to 85.93 mg/100 g. The authors discovered that the dominant tocol fraction is γ-tocopherol, but the content of plastochromanol-8 is also high. Choo et al. [4] and Shim et al. [57] also determined the trace content of α-tocopherol in flaxseed oil. According to Obranović et al. [58], the carotenoid content of cold-pressed flaxseed oil ranges from 0.18 to 0.30 mg/100 g, but the results vary depending on the harvest year and variety. 

### 3.5. Changes in the Content of Phenolic Compounds in Cold-Pressed Flaxseed Oils with the Addition of Ferulic Acid and Its Derivatives in an Oxidation Test

The content of phenolic compounds in cold-pressed flaxseed oil was 1.67 mg/100 g, with phenolic acids accounting for approximately 1% of the total phenolic compounds (Table 5). Among phenolic acids, we found SA (9.87 μg/100 g), FA (3.27 μg/100 g), and p-CA (0.83 μg/100 g). 

It was found that there were noticeable losses in the content of naturally occurring phenolic acids during the oxidation process (Table 5). Losses of FA exceed 10% in oils with the addition of VA and DHFA, losses of SA ranged from more than 50% to nearly 80%, and p-CA was completely degraded in the analyzed oils. Cold-pressed flaxseed oil without additives lost approximately 27% of its total phenolic acid content due to the oxidation process. It was stated that the total phenolic compound content in other oils increased due to the addition of phenolic antioxidants. Despite the addition of individual phenolic antioxidants in an amount of 80 mg/100 g, most of them decayed during the oxidation process. Only about 29% of FA, VA, and DHFA was in the oil samples after the oxidation process, whereas 4-VG losses were even more than 95%.

Siger et al. [59] reported that the total content of phenolic compounds in cold-pressed flaxseed oil ranges from 0.40 to 0.48 mg/100 g. Tańska et al. [3] discovered that the total phenolic compound content can reach as high as 2.19 mg D-catechin/100 g. According to the research of Siger et al. [59], cold-pressed flaxseed oil contains phenolic acids such as p-hydroxybenzoic, VA, and FA, while Hasiewicz-Derkacz et al. [29] found that vanillin is the most abundant phenolic compound in cold-pressed flaxseed oil.

## 4. Conclusions

Research results indicate that ferulic acid and its derivatives differentially affect oxidative stability and the retention of bioactive compounds in cold-pressed flaxseed oil. It seems that not only the type of additive but also the concentration and temperature of the oil treatment are important. The Rancimat tests performed at different temperatures showed that the optimum concentration of the derivatives used was 75 mg/100 g. However, an increasing tendency was observed for ferulic acid. Furthermore, the low-temperature treatment seems to be more favorable for ferulic acid derivatives. In particular, the addition of vanillic acid is not a good antioxidant for oil matrices at accelerated temperatures (>60 °C) because it reduces the induction time of cold-pressed flaxseed oil. The results of the shelf life prediction at 20 °C showed that all tested phenolic additives could be considered effective antioxidants in cold-pressed flaxseed oil for typical culinary applications, but their concentration seems to be important in the development of new oils with increased oxidative stability.

During thermostatic testing, dihydroferulic acid and 4-vinylguaiacol were found to have protective properties for unsaturated fatty acids. A similar tendency was also observed for the content of sterols, while carotenoids and squalene were better protected by ferulic acid (retention of 80%). In contrast, vanillic acid mainly intensified the losses of fatty acids and bioactive compounds, except for carotenoid fractions. The dihydroferulic acid appears to be the most protective against the oil’s main antioxidants, tocopherols (α and γ homologues). In the future, studies on cold-pressed flaxseed oils stored under natural conditions can be carried out to evaluate the potential antioxidant properties of ferulic acid used in combination with dihydroferulic or/and 4-vinylguaiacol.

## Figures and Tables

**Figure 1 foods-12-01088-f001:**
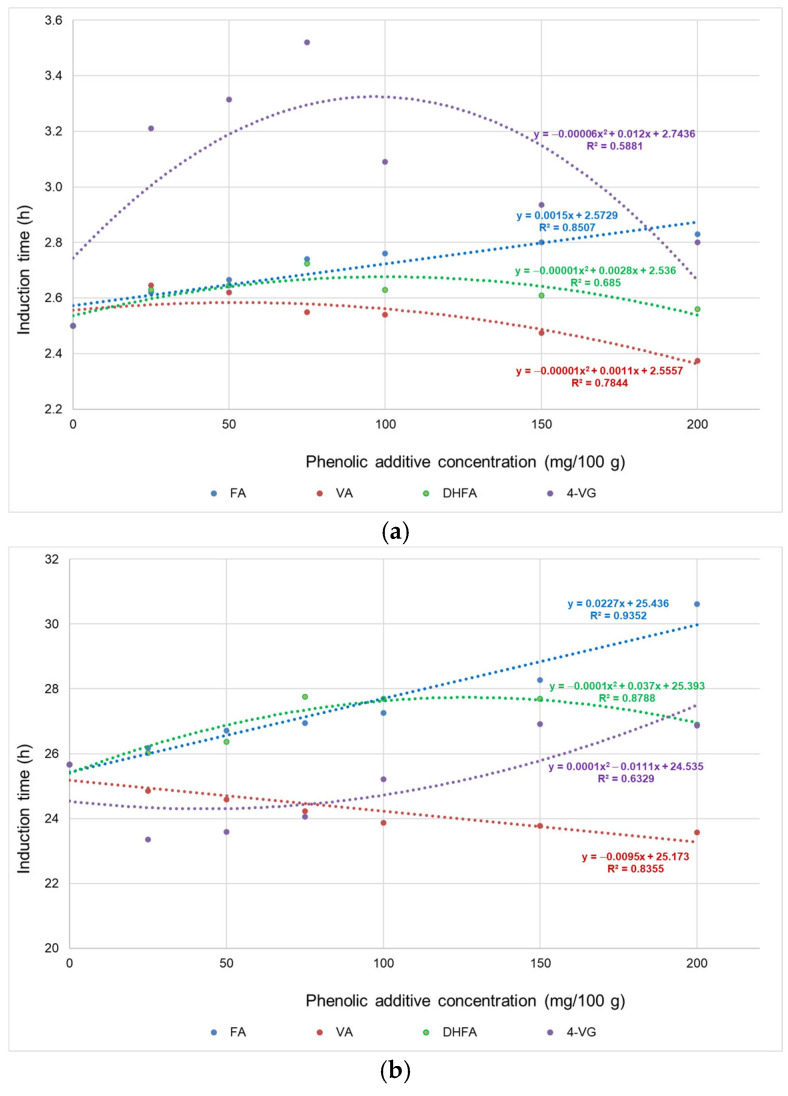
Induction time of cold-pressed flaxseed oil with the addition of phenolic additives (FA—ferulic acid, VA—vanillic acid, DHF—dihydroferulic acid, and 4-VG—4-vinylguaiacol) determined at 110 °C (**a**), 80 °C (**b**), and 60 °C (**c**).

**Table 1 foods-12-01088-t001:** Predicted shelf life ($\bar{x\;}$ ± SD) at 20 °C (months) of cold-pressed flaxseed oil with addition of phenolic additives.

Concentration (mg/100 g)	Phenolic Additive							
	FA	% *	VA	% *	DHFA	% *	4-VG	% *
0	2.47 ^e^ ± 0.11	-	2.47 ^c^ ± 0.11	-	2.47 ^d^ ± 0.11	-	2.47 ^e^ ± 0.11	-
25	3.71 ^d^ ± 0.06	50.2	3.31 ^a^ ± 0.06	34.0	4.62 ^a^ ± 0.09	87.0	2.53 ^de^ ± 0.10	2.4
50	3.84 ^c^ ± 0.10	55.5	3.20 ^a^ ± 0.08	29.5	4.11 ^b^ ± 0.02	66.4	2.69 ^d^ ± 0.08	8.9
75	3.89 ^c^ ± 0.07	57.5	3.13 ^b^ ± 0.06	26.7	3.99 ^c^ ± 0.07	61.5	3.61 ^a^ ± 0.05	46.1
100	3.98 ^c^ ± 0.08	61.1	3.09 ^b^ ± 0.08	25.1	2.55 ^d^ ± 0.08	3.2	3.29 ^b^ ± 0.09	33.2
150	4.39 ^b^ ± 0.05	77.7	3.05 ^b^ ± 0.05	23.5	1.76 ^e^ ± 0.05	−28.7	3.15 ^c^ ± 0.08	27.5
200	5.31 ^a^ ± 0.12	115.0	3.04 ^b^ ± 0.09	23.1	1.63 ^f^ ± 0.04	−34.0	2.06 ^f^ ± 0.05	−16.6

*—percentage change compared to control sample (oil without phenolic additive); FA—ferulic acid; VA—vanillic acid; DHFA—dihydroferulic acid; 4-VG—4-vinylguaiacol; *n* = 3; $\bar{x\;}\;\pm \;\mathrm{SD}—$ mean value ± standard deviation; a–f—mean values in the same column followed by different superscript letters are significantly different (one-way ANOVA and Tukey’s test, *p* ≤ 0.05).

**Table 2 foods-12-01088-t002:** Quality indices ($\bar{x\;}$ ± SD) of fresh cold-pressed flaxseed oil and oil oxidized at 60 °C without and with phenolic additives.

Quality Indices	Fresh Oil	Oxidized Flaxseed Oil with 80 mg/100 g of				
		-	FA	VA	DHFA	4-VG
AV (mg KOH/g)	1.39 ± 0.00 ^b^	1.52 ^a^ ± 0.00	1.67 ^a^ ± 0.00	1.67 ^a^ ± 0.01	1.25 ^b^ ± 0.19	0.83 ^c^ ± 0.00
PV (mEq O_2_/kg)	0.39 ± 0.00 ^e^	6.67 ^d^ ± 0.09	6.30 ^d^ ± 0.33	7.53 ^c^ ± 0.22	8.55 ^b^ ± 0.29	9.88 ^a^ ± 0.08
AnV (-)	0.61 ± 0.00 ^f^	14.33 ^b^ ± 0.06	16.28 ^a^ ± 0.00	2.60 ^e^ ± 0.08	6.91 ^d^ ± 0.02	11.41 ^c^ ± 0.18
Content of Dienes (%)	0.18 ± 0.00 ^e^	0.26 ^d^ ± 0.00	0.30 ^a^ ± 0.00	0.27 ^c^ ± 0.00	0.26 ^d^ ± 0.00	0.29 ^b^ ± 0.00
Content of Trienes (%)	0.00 ± 0.00 ^a^	0.00 ^a^ ± 0.00	0.00 ^a^ ± 0.00	0.00 ^a^ ± 0.00	0.00 ^a^ ± 0.00	0.00 ^a^ ± 0.00

FA—ferulic acid; VA—vanillic acid; DHFA—dihydroferulic acid; 4-VG—4-vinylguaiacol; AV—acid value; PV—peroxide value; AnV—anisidine value; *n* = 3; $\;\bar{x\;}\;\pm \;\mathrm{SD}$—mean value ± standard deviation; a–f—mean values in the same line followed by different superscript letters are significantly different (one-way ANOVA and Tukey’s test, *p* ≤ 0.05).

**Table 3 foods-12-01088-t003:** Fatty acid composition ($\bar{x\;}$ ± SD) of fresh cold-pressed flaxseed oil and oxidized at 60 °C without and with phenolic additives.

Fatty Acids	Fresh Oil	Oxidized Flaxseed Oil with 80 mg/100 g of				
		-	FA	VA	DHFA	4-VG
Palmitic Acid (C16:0) (g/100 g)	4.08 ^b^ ± 0.12	4.08 ^b^ ± 0.06	4.12 ^b^ ± 0.00	3.92 ^b^ ± 0.03	4.28 ^a^ ± 0.09	3.94 ^b^ ± 0.11
Stearic Acid (C18:0) (g/100 g)	1.92 ^a^ ± 0.06	1.37 ^c^ ± 0.28	1.60 ^bc^ ± 0.01	1.93 ^a^ ± 0.10	1.81 ^ab^ ± 0.18	1.92 ^a^ ± 0.07
Oleic Acid (C18:1) (g/100 g)	18.78 ^a^ ± 0.81	17.30 ^a^ ± 0.18	17.90 ^a^ ± 0.38	17.81 ^b^ ± 0.60	18.45 ^a^ ± 0.78	17.57 ^a^ ± 1.27
Linoleic Acid (C18:2) (g/100 g)	12.92 ^a^ ± 0.16	12.75 ^a^ ± 0.01	12.78 ^a^ ± 0.48	11.59 ^b^ ± 0.12	12.90 ^a^ ± 0.20	12.92 ^a^ ± 0.42
α-Linolenic Acid (C18:3) (g/100 g)	60.00 ^a^ ± 1.52	55.90 ^d^ ± 0.13	56.81 ^cd^ ± 1.71	51.13 ^e^ ± 0.24	57.73 ^bc^ ± 0.45	58.36 ^b^ ± 0.08
Total (g/100 g)	97.88 ^a^ ± 0.48	91.40 ^c^ ± 0.38	93.21 ^bc^ ± 2.57	86.37 ^d^ ± 0.90	95.55 ^b^ ± 1.70	94.82 ^bc^ ± 1.75
ΣSFA (g/100 g)	6.18 ^a^ ± 0.06	5.45 ^d^ ± 0.34	5.72 ^cd^ ± 0.01	5.86 ^bc^ ± 0.06	6.09 ^ab^ ± 0.27	5.96 ^b^ ± 0.03
ΣMUFA (g/100 g)	18.78 ^a^ ± 0.81	17.30 ^b^ ± 0.18	17.90 ^ab^ ± 0.38	17.81 ^ab^ ± 0.60	18.45 ^a^ ± 0.78	17.57 ^b^ ± 1.27
ΣPUFA (g/100 g)	72.92 ^a^ ± 1.36	68.65 ^b^ ± 0.14	69.58 ^ab^ ± 2.20	62.71 ^c^ ± 0.36	70.63 ^ab^ ± 0.64	71.28 ^a^ ± 0.51
n-3:n-6	4.64:1	4.38:1	4.44:1	4.41:1	4.47:1	4.52:1

FA—ferulic acid; VA—vanillic acid; DHFA—dihydroferulic acid; 4-VG—4-vinylguaiacol; ΣSFA—sum of saturated fatty acids; ΣMUFA—sum of monosaturated fatty acids; ΣPUFA—sum of polyunsaturated fatty acids; n-3:n-6—ratio of omega 3 to omega 6 fatty acids; *n* = 3; $\bar{x\;}\;\pm \;\mathrm{SD}$—mean value standard deviation; a–d—mean values in the same line followed by different superscript letters are significantly different (one-way ANOVA and Tukey’s test, *p* ≤ 0.05).

**Table 4 foods-12-01088-t004:** Content of lipophilic bioactive compounds (± SD) in fresh cold-pressed flaxseed oil and oxidized at 60 °C without and with phenolic additives.

Bioactive Compounds	Fresh Oil	Oxidized Flaxseed Oil with 80 mg/100 g of				
		-	FA	VA	DHFA	4-VG
Sterols (mg/100 g)						
Campesterol	112.12 ^a^ ± 1.16	105.40 ^c^ ± 0.18	105.75 ^c^ ± 0.95	103.93 ^d^ ± 1.11	107.50 ^b^ ± 0.34	110.52 ^a^ ± 0.18
Stigmasterol	32.05 ^a^ ± 0.99	29.62 ^c^ ± 0.24	30.12 ^bc^ ± 0.55	29.45 ^c^ ± 0.62	30.68 ^ab^ ± 0.15	31.15 ^a^ ± 0.05
β-Sitosterol	216.64 ^a^ ± 1.59	202.41 ^bc^ ± 2.52	204.46 ^b^ ± 5.33	197.62 ^c^ ± 0.24	203.91 ^b^ ± 1.25	210.33 ^a^ ± 1.99
25-Hydroxy-24-Methylcholesterol	53.17 ^a^ ± 0.95	49.78 ^b^ ± 0.51	48.68 ^c^ ± 0.17	48.66 ^c^ ± 0.47	49.53 ^b^ ± 0.43	52.32 ^a^ ± 0.45
Cycloartenol	166.65 ^a^ ± 1.63	153.38 ^b^ ± 1.96	156.75 ^b^ ± 3.86	146.63 ^c^ ± 5.22	156.16 ^bc^ ± 1.21	161.42 ^b^ ± 1.98
Unidentified Sterol-Like Compounds	47.02 ^a^ ± 1.14	46.23 ^a^ ± 0.42	46.67 ^a^ ± 1.95	46.41 ^a^ ± 0.18	47.00 ^a^ ± 0.23	47.02 ^a^ ± 0.02
Total	629.98 ^a^ ± 1.70	586.81 ^c^ ± 4.50	592.42 ^c^ ± 7.47	572.71 ^d^ ± 4.39	594.79 ^c^ ± 3.61	612.75 ^b^ ± 4.63
Tocols (mg/100 g)						
α-Tocopherol	6.14 ^a^ ± 0.24	tr	tr	tr	3.18 ^b^ ± 0.38	tr
γ-Tocopherol	68.18 ^a^ ± 0.67	49.34 ^c^ ± 0.24	50.15 ^c^ ± 1.07	51.81 ^b^ ± 0.68	50.58 ^bc^ ± 0.62	44.83 ^d^ ± 0.08
Plastochromanol-8	32.20 ^a^ ± 0.22	20.87 ^c^ ± 0.15	21.19 ^c^ ± 0.58	22.06 ^b^ ± 0.23	22.05 ^b^ ± 0.09	18.52 ^d^ ± 0.26
Total	106.52 ^a^ ± 1.14	70.21 ^d^ ± 0.39	71.34 ^d^ ± 1.65	73.87 ^c^ ± 0.44	75.81 ^b^ ± 0.91	63.35 ^e^ ± 0.19
Squalene (mg/100 g)						
Total	4.43 ^a^ ± 0.21	3.07 ^c^ ± 0.61	3.52 ^b^ ± 0.63	1.97 ^e^ ± 0.49	2.39 ^d^ ± 0.15	2.50 ^d^ ± 0.05
Carotenoids (mg/100 g)						
α-Carotene	0.07 ^a^ ± 0.02	0.02 ^bc^ ± 0.01	0.03 ^b^ ± 0.01	0.03 ^b^ ± 0.00	0.02 ^bc^ ± 0.00	0.01 ^c^ ± 0.00
β-Carotene	0.32 ^a^ ± 0.05	0.15 ^c^ ± 0.09	0.27 ^ab^ ± 0.10	0.25 ^b^ ± 0.02	0.17 ^c^ ± 0.02	0.06 ^d^ ± 0.01
Lutein	0.32 ^a^ ± 0.00	0.29 ^a^ ± 0.01	0.21 ^b^ ± 0.03	0.21 ^b^ ± 0.06	0.16 ^bc^ ± 0.00	0.10 ^c^ ± 0.01
Zeaxanthin	0.01 ^a^ ± 0.00	0.01 ^a^ ± 0.00	0.01 ^a^ ± 0.00	0.01 ^a^ ± 0.00	0.01 ^a^ ± 0.00	0.01 ^a^ ± 0.00
Total	0.69 ^a^ ± 0.07	0.48 ^b^ ± 0.09	0.55 ^ab^ ± 0.08	0.49 ^b^ ± 0.08	0.35 ^c^ ± 0.02	0.18 ^d^ ± 0.02

FA—ferulic acid; VA—vanillic acid; DHFA—dihydroferulic acid; 4-VG—4-vinylguaiacol; tr—trace amounts; *n* = 3; $\bar{x\;}\;\pm \;\mathrm{SD}$—mean value ± standard deviation; a–e—mean values in the same line followed by different superscript letters are significantly different (one-way ANOVA and Tukey’s test, *p* ≤ 0.05).

**Table 5 foods-12-01088-t005:** Content of phenolic compounds ($\bar{x\;}\;$± SD) in fresh cold-pressed flaxseed oil and oxidized oil at 60 °C without and with phenolic additives.

Phenolic Compounds	Fresh Oil	Oxidized Flaxseed Oil with 80 mg/100 g of				
		-	FA	VA	DHFA	4-VG
p-CA (μg/100 g)	0.83 ^a^ ± 0.00	0.63 ^b^ ± 0.00	0.00 ^c^ ± 0.00	0.00 ^c^ ± 0.00	0.00 ^c^ ± 0.00	0.00 ^c^ ± 0.00
FA (μg/100 g)	3.27 ^b^ ± 0.03	3.17 ^b^ ± 0.12	9909.25 ^a^ ± 86.24	2.96 ^b^ ± 0.17	2.92 ^b^ ± 0.31	3.17 ^b^ ± 0.02
SA (μg/100 g)	9.87 ^a^ ± 0.10	6.33 ^b^ ± 0.21	2.45 ^e^ ± 0.24	3.90 ^d^ ± 0.00	4.43 ^cd^ ± 0.40	4.91 ^c^ ± 0.34
VA (μg/100 g)	0.00 ^b^ ± 0.00	0.00 ^b^ ± 0.00	0.00 ^b^ ± 0.00	19,877.08 ^a^ ± 31.75	0.00 ^b^ ± 0.00	0.00 ^b^ ± 0.00
DHFA (μg/100 g)	0.00 ^b^ ± 0.00	0.00 ^b^ ± 0.00	0.00 ^b^ ± 0.00	0.00 ^b^ ± 0.00	1579.82 ^a^ ± 20.44	0.00 ^b^ ± 0.00
4-VG (μg/100g)	0.00 ^b^ ± 0.00	0.00 ^b^ ± 0.00	0.00 ^b^ ± 0.00	0.00 ^b^ ± 0.00	0.00 ^b^ ± 0.00	90.20 ^a^ ± 1.56
Total Phenolic Compounds (mg/100 g)	1.67 ^d^ ± 0.02	1.15 ^e^ ± 0.02	23.42 ^a^ ± 0.58	22.17 ^b^ ± 0.02	23.49 ^a^ ± 0.08	3.19 ^c^ ± 0.13

p-CA—p-Coumaric acid; FA—ferulic acid; SA—sinapic acid; VA—vanillic acid; DHFA—dihydroferulic acid; 4-VG—4-vinylguaiacol; *n* = 3; $\bar{x\;}\pm \mathrm{SD}\;$—mean value ± standard deviation; a–e—mean values in the same line followed by different superscript letters are significantly different (one-way ANOVA and Tukey’s test, *p* ≤ 0.05).

## Data Availability

The data presented in this study are available on request from the corresponding author.

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
