# Peer review of "Effect of Ferulic Acid and Its Derivatives on Cold-Pressed Flaxseed Oil Oxidative Stability and Bioactive Compounds Retention during Oxidation"

_foods, 2023, doi:10.3390/foods12051088_

Round 1
Reviewer 1 Report
In this paper, the effect of FA and its esters on the cold pressed flax seed oil was investigated.
It is interesting. However, there are some issues should be improved.
1. Please give the information about the processing of flax seed oil. It was obtained using cold pressing. How about the process conditions?
2. Line 157, please use the international official methods, for example, AOCS methods and IUPAC methods.
3. What is “IT” in line 271? In line 147, it is IP?
4. The antioxidant activity of FA is relative to its solubility and concentration in the oil. And there are some relative reports about the improvement of FA solubility (DOI 10.1002/aocs.12255, DOI 10.1039/d2ra01683d, DOI 10.1002/aocs.12047, DOI, 10.1021/acs.jafc.5b03479 ) should be included.
5. Please explain the results of “The oil retained only approximately 29% of added phenolic compounds such as FA, VA, and DHFA, while 4-VG losses were even more than 95%”?
6. Could you explain the synergistic interaction between phenolic compounds and tocols?
Author Response
We would like to thank the Reviewers and the Editor for the time taken to review our manuscript entitled “Effect of Ferulic Acid and Its Derivatives on Cold-Pressed Flaxseed Oil Oxidative Stability and Bioactive Compounds Retention during Oxidation”, and the constructive criticism and modification suggestions. The remarks and comments of the Reviewers were valuable and helped us to improve the text quality. The answers for the comments of the Reviewer 1 are presented below. Additional modifications to overall text were conducted to improve language.
Response to #Reviewer 1 Comments
Point 1: Please give the information about the processing of flax seed oil. It was obtained using cold pressing. How about the process conditions?
Response 1: As mentioned in Materials and Methods (section 2.2), the cold-pressed flaxseed oil used in the research was commercial, and it was purchased from the company immediately after its production. Unfortunately, we do not have additional information about the process conditions due to the fact that it is covered by a company secret. In addition, there is no obligation to provide information about production conditions on the package label. However, we decided to include some details about the production of a conventional flaxseed oil in ‘Introduction’ (section 1).
Point 2: Line 157, please use the international official methods, for example, AOCS methods and IUPAC methods.
Response 2: It has been corrected according to the Reviewer’s suggestion. The AOCS methods have been added to ‘Materials and Methods’ (sections 2.3 and 2.4) and ‘References’.
Point 3: What is “IT” in line 271? In line 147, it is IP?
Response 3: We thank the Reviewer for pointing out this error. We standardized the reporting of Rancimat test results in the text using the abbreviation IT defined in ‘Materials and Methods’ (section 2.3).
Point 4: The antioxidant activity of FA is relative to its solubility and concentration in the oil. And there are some relative reports about the improvement of FA solubility (DOI 10.1002/aocs.12255, DOI 10.1039/d2ra01683d, DOI 10.1002/aocs.12047, DOI, 10.1021/acs.jafc.5b03479) should be included.
Response 4: We thank the Reviewer for indicating the direction of increasing the substantive content of our manuscript. As suggested, we added 3 out of 4 articles and cited them in the revised manuscript.
Point 5: Please explain the results of “The oil retained only approximately 29% of added phenolic compounds such as FA, VA, and DHFA, while 4-VG losses were even more than 95%”?
Response 5: Phenolic additives were added to the oil in an amount of 80 mg/100 g. Accordingly, the initial content of the phenolic compounds in the oils with additives was assumed to be 81.67 mg/100 g in total. Thus, the total phenolic content was 23.42 mg/100 g in oil with the addition of ferulic acid, 22.17 mg/100 g in oil with the addition of vanillic acid, and 23.49 mg/100 g in oil with the addition of dihydroferulic acid (Table 5). These values indicate the loss of individual additives in the amount of approx. 71-73%. In turn, the content of 4-vinylguiacol in the oil with its addition was only 3.19 mg/100 g after the oxidation process (Table 5), which indicates a loss of more than 95% of this compound.
Point 6: Could you explain the synergistic interaction between phenolic compounds and tocols?
Response 6: This sentence was only loosely connected with the manuscript subject. Finally, we decided to delete it (without prejudice to the quality of the manuscript).
We believe that present form of our manuscripts is much clearer, concise and valuable. And we hope that our corrections and answers will be satisfying for the Reviewers and the Editor.

Reviewer 2 Report
This manuscript is written well. The authors provided details in the Discussion par. However, a few major issues need to solve before publication.
The introduction is lengthy. Please remove too general sentences and rearrange the text.
The authors need to show the oxidation stability performance of each sample.
Insert references in results and discussion in each finding and conclusion.
Please compare your study with recent studies.
Insert the following references in the text https://doi.org/10.1016/j.focha.2022.100016, International Journal of Biological Macromolecules 171 (2021) 457-464; and International Journal of Biological Macromolecules 136 (2019) 661-667
The conclusion section need to be shortened. Please include the main finding of this study. Make it maximum two paragraphs.
The authors have not studied anything related to bioactive compounds please comment on it in the manuscript.
Author Response
We would like to thank the Reviewers and the Editor for the time taken to review our manuscript entitled “Effect of Ferulic Acid and Its Derivatives on Cold-Pressed Flaxseed Oil Oxidative Stability and Bioactive Compounds Retention during Oxidation”, and the constructive criticism and modification suggestions. The remarks and comments of the Reviewers were valuable and helped us to improve the text quality. The answers for the comments of the Reviewer 2 are presented below. Additional modifications to overall text were conducted to improve language.
Response to #Reviewer 2 Comments
Point 1: The introduction is lengthy. Please remove too general sentences and rearrange the text.
Response 1: Some general sentences (e.g., effect of storage on oil deterioration; occurring of ferulic acid forms in plants; effect of phenolic compounds on the oil oxidative stability studied by other researchers) have been removed from ‘Introduction’ section according to the Reviewer’s suggestion.
Point 2: The authors need to show the oxidation stability performance of each sample.
Response 2: The induction period of the tested oils was determined using the Rancimat test at 60, 80, and 110 °C, and the results were statistically processed as the mean values. Since mean values are presented and used in the development of Figure 1, we believe that there is no need to present and repeat specific values. Moreover, according to the information contained in the back matter in Data Availability Statement, the reader can obtain more information about the research and its results by sending a query to the corresponding author.
Point 3: Insert references in results and discussion in each finding and conclusion.
Response 3: It has been corrected according to the Reviewer’s suggestion.
Point 4: Please compare your study with recent studies.
Response 4: A discussion about the previous studies and updated research on the same topics have been added.
Point 5: Insert the following references in the text https://doi.org/10.1016/j.focha.2022.100016, International Journal of Biological Macromolecules 171 (2021) 457-464; and International Journal of Biological Macromolecules 136 (2019) 661-667.
Response 5: We thank the Reviewer for indicating the direction of increasing the substantive content of our manuscript. As suggested, we added 2 out of 3 articles and cited them in the manuscript.
Point 6: The conclusion section need to be shortened. Please include the main finding of this study. Make it maximum two paragraphs.
Response 6: It has been corrected according to the Reviewer’s suggestion.
Point 7: The authors have not studied anything related to bioactive compounds please comment on it in the manuscript.
Response 7: Our research is based not only on showing changes in the oxidative stability of cold-pressed flaxseed oil after the addition of phenolic additives but also on showing changes in bioactive compounds in this oil during oxidation. This information is included in the title of the manuscript (“Effect of Ferulic Acid and Its Derivatives on Cold-Pressed Flaxseed Oil Oxidative Stability and Bioactive Compounds Retention during Oxidation”), and changes in the content of bioactive compounds are presented in ‘Results and Discussion’ (section 3.4).
We believe that present form of our manuscripts is much clearer, concise and valuable. And we hope that our corrections and answers will be satisfying for the Reviewers and the Editor.

Reviewer 3 Report
Comments and Suggestions for Authors
1. -Please add the unit of fatty acid composition in Table 3.
- Line 403-408 : Please check and correct the unit of total fatty acid (line 403) and palmitic acid (line 408)
2. -Please add the unit of phenolic compound in Table 5.
- Line 523 : sinapic (9.87 μg 100 g) and ferulic acids (3.27 μg/100 g) Please check it is correct because it is different from the data in Table 5.
- Line 524-535 : very confusing please rewrite and make it clearer
: The use of phenolic additives increased the loss of naturally occurring phenolic acids in cold-pressed flaxseed oil. What is this sentence means?
: p-CA was completely degraded in the analyzed oils. Is only p-CA ??
: line 527 Losses of FA reached over 10% in oils with the addition of VA and DHFA - Not understand why over 10% because in Table 5 FA content showed only 0.00 in -, VA, DHFA and 4-VG
: line 528 losses of SA were the highest of all – not understand please explain.
: Line 529 while losses in other oils ranged from over 50% to almost 80% --where is 50% and 80% come from because in Table 5 SA of FA VA DHFA showed the same number 0.00.
: The oil retained only approximately 29% of added phenolic compounds such as FA, VA, and DHFA ----please explain 29% come from??
3. conclusion should be rewritten to make it more concise.
4. Numerous grammatical improvements are needed throughout the manuscript.

Author Response
We would like to thank the Reviewers and the Editor for the time taken to review our manuscript entitled “Effect of Ferulic Acid and Its Derivatives on Cold-Pressed Flaxseed Oil Oxidative Stability and Bioactive Compounds Retention during Oxidation”, and the constructive criticism and modification suggestions. The remarks and comments of the Reviewers were valuable and helped us to improve the text quality. The answers for the comments of the Reviewer 3 are presented below. Additional modifications to overall text were conducted to improve language.
Response to #Reviewer 3 Comments
Point 1: Please add the unit of fatty acid composition in Table 3.
Response 1: The unit of fatty acids in oils is defined in ‘Materials and Methods’ (section 2.4.). As suggested by the Reviewer, the unit of fatty acids was added as g/100 g in Table 3.
Point 2: Line 403-408: Please check and correct the unit of total fatty acid (line 403) and palmitic acid (line 408).
Response 2: The sentences have been checked and corrected as suggested by the Reviewer. The sentence with the content of palmitic acid had an incorrect unit, which was corrected from mg/100 g to g/100 g.
Point 3: Please add the unit of phenolic compound in Table 5.
Response 3: As suggested by the Reviewer, the unit of phenolic compounds content was added as mg/100 g (total phenolic compounds) or mg/100 g (individual phenolic compounds) in Table 5.
Point 4: Line 523: sinapic (9.87 μg 100 g) and ferulic acids (3.27 μg/100 g) Please check it is correct because it is different from the data in Table 5.
Response 4: Thanks to the Reviewer for pointing out this error we overlooked. The results of the phenolic acids content in Table 5 should be expressed in mg/100 g. It has been corrected in the revised manuscript, and now the sentence is consistent with the data given in Table 5.
Point 5: Line 524-535: very confusing please rewrite and make it clearer.
Response 5: The discussion of the content of phenolic compounds in oils has been modified. We believe that the description is easier to read and is consistent with the data given in Table 5.
Point 6: The use of phenolic additives increased the loss of naturally occurring phenolic acids in cold-pressed flaxseed oil. What is this sentence means?
Response 6: The questionable sentence has been deleted.
Point 7: p-CA was completely degraded in the analyzed oils. Is only p-CA??
Response 7: The results of the phenolic acids content have been improved in Table 5. Therefore, the discussion of the content of phenolic compounds in oils has been modified.
Point 8: Line 527 Losses of FA reached over 10% in oils with the addition of VA and DHFA - Not understand why over 10% because in Table 5 FA content showed only 0.00 in -, VA, DHFA and 4-VG.
Response 8: We thank the Reviewer for pointing out this error. The entire discussion of the content of phenolic compounds in oils has been improved, and now is consistent with the data in Table 5.
Point 9: Line 528 losses of SA were the highest of all – not understand please explain.
Response 9: This sentence has been removed from the text because it misled the reader.
Point 10: Line 529 while losses in other oils ranged from over 50% to almost 80% - where is 50% and 80% come from because in Table 5 SA of FA VA DHFA showed the same number 0.00.
Response 10: The entire discussion of the content of phenolic compounds in oils has been improved and is now consistent with the data in Table 5.
Point 11: The oil retained only approximately 29% of added phenolic compounds such as FA, VA, and DHFA ----please explain 29% come from??
Response 11: Phenolic additives were added to the oil in an amount of 80 mg/100 g. Accordingly, the initial content of the phenolic compounds in the oils with additives was assumed to be 81.67 mg/100 g in total. Thus, the total phenolic content was 23.42 mg/100 g in oil with the addition of ferulic acid, 22.17 mg/100 g in oil with the addition of vanillic acid, and 23.49 mg/100 g in oil with the addition of dihydroferulic acid (Table 5). These values indicate the loss of individual additives in the amount of approx. 71-73%.
Point 12: Conclusion should be rewritten to make it more concise.
Response 12: ‘Conclusion’ section has been shortened, and the paper's most important findings, including the future trends of the topic have been summarized.
Point 13: Numerous grammatical improvements are needed throughout the manuscript.
Response 13: We regret there were problems with the English. The paper has been carefully revised by a native English speaker to improve the grammar and readability.
We believe that present form of our manuscripts is much clearer, concise and valuable. And we hope that our corrections and answers will be satisfying for the Reviewers and the Editor.

Round 2
Reviewer 2 Report
Authors improved the mnauscript. It is hard to find revision part in the manuscript as it is not highlight.